# The Role and Place of Antioxidants in the Treatment of Male Infertility Caused by Varicocele

**DOI:** 10.3390/jcm11216391

**Published:** 2022-10-28

**Authors:** Marek Szymański, Piotr Domaracki, Angelika Szymańska, Tomasz Wandtke, Robert Szyca, Łukasz Brycht, Karolina Wasilow, Wojciech Jan Skorupski

**Affiliations:** 1Department of Women’s Health and Reproductive Medicine, Collegium Medicum in Bydgoszcz, Nicolaus Copernicus University in Toruń, 85-067 Bydgoszcz, Poland; 2NZOZ Medical Center, Clinic of Infertility Treatment “Genesis”, 85-435 Bydgoszcz, Poland; 3Clinic of Gynaecology and Oncological Gynecology, 10th Military Research Hospital and Polyclinic, IPHC, 85-681 Bydgoszcz, Poland; 4Department of Lung Diseases, Neoplasms and Tuberculosis, Faculty of Medicine, Nicolaus Copernicus University in Toruń, 85-094 Bydgoszcz, Poland; 5Clinic of Surgery and Oncological Surgery, 10th Military Research Hospital and Polyclinic, IPHC, 85-681 Bydgoszcz, Poland; 61st Department of Cardiology, Poznan University of Medical Sciences, 61-848 Poznan, Poland

**Keywords:** varicocele, male infertility, antioxidant therapy, oxidative stress

## Abstract

The inability to become pregnant for at least 1 year despite regular unprotected intercourse may indicate infertility of one or both partners. This problem affects approximately 10–20% of couples worldwide, regardless of race, with male infertility reported to account for 25–60% of cases. Among the most common pathological causes of male infertility is the presence of varicocele and chronic infections of the male reproductive system. This study was performed using data collected at the Genesis Infertility Treatment Clinic, Bydgoszcz, Poland, between 1 January 2015 and 30 June 2017. A total of 163 men meeting the inclusion criteria were selected and divided into the idiopathic infertility group (78 men) and varicocele-related infertility group (85 men). All patients received treatment with a male fertility supplement containing a combination of 1725 mg of L-carnitine fumarate, 500 mg of acetyl-L-carnitine, 90 mg of vitamin C, 20 mg of coenzyme Q10, 10 mg of zinc, 200 µg of folic acid, 50 µg of selenium, and 1.5 µg of vitamin B12 (Proxeed^®^ Plus, Sigma-Tau, Italy) twice a day for a period of 6 months from the time of the diagnosis of infertility. The treatment resulted in significant improvements in general semen parameters, particularly sperm count, sperm concentration, total motility, and progressive motility. This antioxidant therapy produced a particularly marked therapeutic benefit in patients with Grade III varicocele, with a greater improvement in progressive motility than in men with less severe or no varicocele. The use of the antioxidant preparation examined here seems reasonable in men with idiopathic infertility and as an adjuvant in those with varicocele-related infertility in whom surgical treatment has resulted in no improvement. Its use should be considered particularly in patients with Grade III varicocele who do not wish to undergo surgical treatment or in whom such a treatment is not possible for various reasons.

## 1. Introduction

The inability to become pregnant for at least 1 year despite regular intercourse without contraception may indicate infertility in one or both partners [1,2]. This problem affects approximately 10–20% of couples worldwide, regardless of race, and male infertility accounts for between 25% and 60% of cases [3,4,5].

### 1.1. Medical and Environmental Factors and Male Infertility

A number of environmental and medical/health-related factors may be involved in the deterioration of quantitative and qualitative semen parameters. The most commonly reported environmental factors include the adverse effects of stress; tobacco; alcohol and drug abuse; being overweight; having an unhealthy diet; genital trauma; an exposure to toxic substances such as heavy metals, pesticides, and xenoestrogens that disrupt homeostasis of the pituitary–hypothalamic–gonadal axis; and an exposure to heat and electromagnetic radiation [6,7]. The most common pathological causes of male infertility include the presence of varicocele and chronic infections in the male reproductive system [8]. Such factors can lead to the production of oxygen free radicals and the progression of oxidative stress, which is considered to underlie male infertility [9,10,11].

Male infertility is idiopathic in about 30% of cases, where there is no obvious cause of the decrease in semen quality and/or quantity. However, such patients frequently also show elevated levels of oxidative stress [12].

### 1.2. Varicocele of the Vas Deferens

Varicocele is one of causes of the declined sperm quality and low sperm production, which can lead to infertility in males. Varicoceles occur in approximately 5–22% of the male population (most commonly in the 15–25 age group) and account for nearly 20–42% of cases of male infertility, representing one of the most common detectable causes of infertility in men [13]. However, not every occurrence of varicocele is associated with fertility disorders, with nearly 80% of men diagnosed with varicocele retaining the normal general semen parameters [14]. Varicocele is caused by the elongation and dilation of the pampiniform venous plexus in the testicle. Such lesions are significantly more common on the left than the right side [15]. There are several experimental and epidemiological findings which support the idea that inflammatory mechanisms play an essential role in varicocele pathogenesis [16,17]. Varicocele of the seminal cord is classified according to the degree of severity: grade I–IV, [18]. Although the precise causes of fertility disturbances in some patients with varicocele have not been elucidated, a decreased reproductive potential has been suggested to be related to several factors, including an increased temperature in the testicular region due to an increased blood flow, which causes a decrease in the activities of the antioxidant enzymes superoxide dismutase (SOD) and catalase in sperm. This in turn results in an increased hydrogen peroxide (H2O2) concentration, reflux of renal and adrenal toxic metabolites, reduced blood supply to the testes and associated hypoxia and subsequent tissue damage, hormonal abnormalities, autoimmune diseases, and the production of oxygen free radicals [19,20]. Oxidative stress has been widely reported to be the main cause of tissue damage and decreased semen quality and quantity in patients with varicocele [14,21,22]. The seminal fluid of fertility-impaired men with a positive diagnosis of varicocele has significantly higher levels of free radicals than men without such lesions [23]. The reduced reproductive potential in men with a positive diagnosis of varicocele is correlated with the elevated levels of oxidative stress, reduced total sperm count, reduced sperm motility, poor sperm morphology, elevated fragmentation of mitochondrial and genomic DNA (up to 8× that of the healthy population), increased frequencies of sperm apoptosis and necrosis, and smaller testis size on the side with varicocele [18,24]. Varicocele and smoking (10/day) have synergistic adverse effects on sperm motility and morphology [25]. At present, the standard treatment for patients with established varicocele, including men with fertility disorders, is surgical removal [14,26]. Such a treatment has been shown to reduce heat stress in the testicular region and improve or even normalize the activity of the protective enzymatic and non-enzymatic antioxidant systems in male semen. However, data on pregnancy rates following treatment are not conclusive [27]. Unfortunately, some men undergoing treatment for varicocele-related infertility do not show improvement in semen parameters [28]. Alternative, noninvasive therapeutic interventions involving the use of oxidative stress-lowering agents may be beneficial in these, and possibly other, patients [29]. However, the results of the comparative analyses of surgical treatment alone and surgical treatment combined with antioxidant supplementation have been inconclusive [29,30].

### 1.3. Treatment with Antioxidants

Antioxidants are commonly used for the treatment of male infertility with increased levels of oxidative stress, including patients with varicocele and those with idiopathic infertility [8]. The main mechanism of the action of such preparations involves the prevention of the peroxidation of membrane lipids, and to a lesser extent DNA [28]. The rationale behind oral antioxidants intake and positive effects on male reproduction outcome is only supported by a few studies. [31] Treatment with such agents, particularly in patients with varicocele, may contribute not only to the improvement of general semen parameters, but also to a reduction in the degree of DNA fragmentation in sperm with a consequent increase in fertility [29]. Although there are some contrasting reports, the oral consumption of compounds with an antioxidant activity appears to improve the sperm parameters, such as motility and concentration, and decrease DNA damage, but there is not sufficient evidence that fertility rates and live birth really improve after antioxidants intake. Moreover, it depends on the type of antioxidants, treatment duration, and even the diagnostics of the man’s fertility, among other factors [31]. In this study we present our clinical results.

## 2. Materials and Methods

This retrospective study was performed using medical information collected at the Genesis Infertility Treatment Clinic, Bydgoszcz, Poland, between 1 January 2015 and 30 June 2017. The study was approved by the Local Bioethics Committee of Nicolaus Copernicus University in Toruń; Ludwik Rydygier Collegium Medicum in Bydgoszcz, Poland (approval number: KB564/2017). The study was not sponsored by a pharmaceutical company or any drug marketing agency.

A group of 257 men in whom diagnostic procedures were carried out to determine the cause of infertility were initially selected for inclusion in the study. Ninety-four patients were excluded from further analysis because they did not meet the inclusion criteria. The remaining 163 men meeting the inclusion criteria were divided into the idiopathic infertility group (78 men) and varicocele-related infertility group (85 men) according to the criteria outlined below.

Patients with idiopathic infertility were selected by excluding those in whom diagnostic procedures (i.e., tests for blood levels of the luteinizing hormone [LH], follicle-stimulating hormone [FSH], and testosterone, and the direct examination of the genital area, including the Doppler ultrasound) confirmed the presence of any detectable cause of infertility, particularly varicocele, hormonal disorders, and infections. The idiopathic infertility group consisted of 78 men with a leukocyte content in semen < 1 million/mL and fulfilling one or more of the following inclusion criteria: sperm count < 39 million/ejaculate (or sperm concentration in semen < 15 million/mL), progressive motility < 32%, and presence of morphologically pathological sperm forms in semen > 96%.

Patients with varicocele-related infertility were selected by excluding those in whom diagnostic procedures (i.e., the direct examination of the genital area in accordance with routine clinical work-up, including the Doppler ultrasound) confirmed the absence of varicocele. Patients with concomitant varicocele and hormonal disorders (based on the analysis of blood LH, FSH, and testosterone levels) and/or an infection of the reproductive or urinary tract were also excluded. The varicocele-related infertility group consisted of 85 patients characterized by a semen leukocyte content < 1 million/mL and one or more of the following inclusion criteria: sperm count < 39 million/ejaculate (or sperm concentration in semen < 15 million/mL), progressive motility < 32%, and presence of morphologically pathological sperm forms in semen > 96%. The varicocele-related infertility group was further divided into three subgroups according to the stage of varicocele: Grade I (48 patients), Grade II (25 patients), and Grade III (12 patients).

### 2.1. Antioxidant Therapy

All patients received treatment with an antioxidant male fertility supplement containing L-carnitine fumarate (1725 mg), acetyl-L-carnitine (500 mg), ascorbic acid (vitamin C, 90 mg), coenzyme Q10 (20 mg), zinc (10 mg), folic acid (200 μg), selenium (50 μg), and vitamin B12 (1.5 μg) (Proxeed Plus; Sigma-Tau, Pomezia, Rome, Italy) twice a day for a period of 6 months from the time of the diagnosis of infertility.

### 2.2. Semen Testing

The basic seminal parameters were evaluated by a European Society of Human Reproduction and Embryology (ESHRE)-certified embryologist following the fifth edition of the World Health Organization (2010) guidelines.

The following semen parameters were analyzed at the baseline and after 6 months of treatment with the supplement: the ejaculate volume (mL), total sperm count (million/ejaculate), sperm concentration (million/mL), progressive motility (including fast and slow A and B forms, respectively) and total sperm motility (% and million/mL), pathological form content (%), and semen leukocyte concentration (million/mL). The semen was evaluated by one European Society of Human Reproduction and Embryology (ESHRE)-certified clinical embryologist according to the WHO reference values (5th edition) [2]. Semen, after 2–5 days of sexual abstinence, was donated by patients into sterile containers. Semen analysis took place using a Makler chamber. A Makler sperm counting chamber (Sefi Medical Instruments, Haifa, Israel) inserted into a light microscope (Carl Zeiss Jena, Jena, Germany) with a Ph2 phase contrast was used for the microscopic evaluation of the semen. A small drop of semen, well mixed by pipetting, was placed in the center of a Makler chamber, and the chamber was covered with a coverslip. The preparations were then viewed under a magnification of 200× at room temperature. The type of movement was then assessed, distinguishing among fast (type A) and slow (type B) forward (type A) and slow (type B) sperm, non-progressive (type C) sperm (local motility), and non-moving (type D) sperm (no visible motility).

### 2.3. Statistical Analysis

Statistical analyses were performed using Microsoft Excel 2013 (Microsoft, Redmond, WA, USA) and Statistica 13 (StatSoft, Tulsa, OK, USA). The Shapiro–Wilk test confirmed the non-normality of the data distribution, so non-parametric tests were used for the further analyses. The Friedman test was used for the comparisons of the results between the baseline and after 6 months of treatment and the Mann–Whitney U test was used for making comparisons between the groups with and without varicocele. The initial and final results (Table 1, Table 2, Table 3 and Table 4) were analyzed using the Friedman test to verify the significance of the observed changes after the treatment. The analyses were performed on the greatest changes in the initial parameters at month 6 of the treatment within the whole study population of 163 patients. The effects of the presence of varicocele on the semen parameters were examined using the Mann–Whitney U test (Table 5, Table 6, Table 7 and Table 8). In all analyses, *p* < 0.05 was taken to indicate a statistical significance.

## 3. Results

### 3.1. Evaluation of the Effects of Antioxidant Therapy on General Semen Parameters

There were statistically significant changes in the semen quality and/or quantity parameters after 6 months of antioxidant therapy with Proxeed Plus (Sigma-Tau) compared to the baseline. With the exception of the percentage of pathological forms (Table 1) and the percentage, concentration, as well as quantity of D forms of spermatozoa (Table 2, Table 3 and Table 4), which showed no significant changes after the antioxidant therapy, significant increases were observed in all the semen quality and/or quantity parameters after the treatment (Table 1, Table 2, Table 3 and Table 4). These results confirm that Proxeed Plus had a positive effect on the semen quality. Following a treatment with Proxeed Plus, significant increases were observed in the volume of ejaculate in 42% of patients, the sperm concentration in 85%, and the total sperm count in 81% (Table 1). Improvements in the total and progressive sperm motility (i.e., increases in scores) were observed in 55% and 66% of patients, respectively.

Analysis of the sperm motility (A and B forms, fast and slow progressive motility, respectively; C, local motility; D, no visible motility) showed increases in the percentages of A and B forms after the treatment in 54% and 74% of patients, respectively; the concentrations of A and B forms in 66% and 77%, respectively; and in the total sperm count in the ejaculate in 71% and 80%, respectively (Table 2, Table 3 and Table 4). The analysis also showed a statistically significant increase after therapy in the percentage of locally motile forms (C form) in 50% of patients, which was most likely due to the absolute increase in the total sperm number in the semen. At the same time, higher rates of decreases in the occurrence of locally motile C form spermatozoa in semen in relation to the A and B forms showing a progressive motility were noted in terms of a percentage (31% vs. 21% vs. 18%, respectively), concentration (19% vs. 4% vs. 6%, respectively) and absolute total numbers (20% vs. 7% vs. 19%, respectively). There were no significant differences after therapy in the percentage, concentration, or absolute total numbers of immobile D form sperm. As in the case of locally motile C form sperm, higher rates of decreases in the percentage, concentration, and total number of D form sperm were observed compared to those in A, B, and C forms.

Despite the significant positive effects of antioxidant therapy, some patients showed a lack of improvement or even worsening of the semen quantity/quality parameters. Next, we examined whether the presence of varicocele or other idiopathic factors responsible for male infertility may be associated with a weaker response to the antioxidant therapy, and whether the supplement used would have a particularly beneficial effect in either of the two groups of patients.

### 3.2. Effects of the Presence of Varicocele on Semen Parameters at Baseline and Differences after Treatment

The total patient population (n = 163) consisted of 85 men with varicocele (Grades I, II, and III; varicocele-related infertility group) and 78 men without varicocele (idiopathic infertility group). The general semen parameters are shown in Table 5.

There were no significant differences in the baseline values before therapy between the two groups with the exception of a progressive mobility, which was significantly reduced in the varicocele-related infertility group compared to the idiopathic infertility group (mean: 40.8% ± 33.2% vs. 53.8% ± 37.2%, respectively; median: 33.3% vs. 57.1%, respectively). Poorer values of individual general semen parameters were associated with an increasing grade of varicocele in the varicocele-related infertility group. Patients with Grade III varicocele showed a significantly lower sperm concentration and absolute total number and higher percentage of pathological forms compared to the idiopathic infertility group (0.6 ± 0.4 million/mL vs. 6.9 ± 10.6 million/mL, respectively, *p* < 0.001; 1.7 ± 1.3 million/ejaculate vs. 25.4 ± 47.3 million/ejaculate, respectively, *p* < 0.001; 98.4% ± 1.4% vs. 96.6% ± 1.2%, respectively, *p* = 0.015) and corresponding values of those with lower grade varicocele (Grade I: 9.3 ± 11 million/mL, *p* < 0.001; 30.6 ± 37.2 million/ejaculate, *p* < 0.001; 96.4% ± 1.3%, *p* = 0.005, respectively; Grade II: 4.9 ± 4.7 million/mL, *p* = 0.018; 18.7 ± 18.9 million/ejaculate, *p* = 0.006; 95.8% ± 1.5%, *p* < 0.001, respectively). There were no significant differences in the changes from the baseline in the values of the general semen parameters after antioxidant therapy between the groups. In the varicocele-related infertility group, the greatest improvements were seen in the Grade III varicocele subgroup. After treatment, the percentage of motile forms was significantly greater in that subgroup than in the Grade II subgroup (32.8% ± 25.4% vs. 10.3% ± 22.6%, respectively, *p* = 0.029), including those with a progressive motility, which were significantly greater in the Grade III subgroup than in the Grade I and Grade II subgroups. 

Table 6, Table 7 and Table 8 shows the data on spermatozoa in semen divided into subpopulations according to the differences in motility before and after therapy in the following groups: progressive (A and B forms, fast and slow, respectively), local (C form), and immotile (D form). Prior to therapy, there were no significant differences among any groups. However, an extended statistical analysis taking into account the severity of varicocele before treatment indicated significant differences in the total number and concentration of A form spermatozoa in the Grade III varicocele subgroup compared to the Grade I subgroup. These observations suggest that Grade III varicocele is associated with significantly poorer values of the general semen parameters.

Similarly to the baseline observations, there were no significant differences among groups after therapy (Table 6, Table 7 and Table 8). However, according to the severity, the Grade III group showed a significantly greater improvement in the percentage of B form sperm after therapy compared to the idiopathic infertility group (28.7% ± 16.6% vs. 10.8% ± 22%, respectively, *p* = 0.015), Grade I subgroup (11.9% ± 18.8%, *p* = 0.045), and Grade II subgroup (8.4% ± 23.4%, *p* = 0.048).

## 4. Discussion

Varicocele is a common condition, with an estimated incidence of up to one in five men [32]. However, the precise relationship between the presence of varicocele and the deterioration of qualitative and quantitative semen parameters is still not clear. This is all the more controversial because some men diagnosed with seminal varicocele do not have fertility disorders [33,34,35]. The main factor related to a reduced reproductive potential associated with the presence of varicocele is oxidative stress, which affects the process of spermatogenesis as well as inducing direct functional and structural damage to male germ cells, including effects on the sperm cell membrane and damage to genomic and mitochondrial DNA [36]. The evidence regarding the role of L-carnitine as a primary or adjuvant treatment of varicocele is sparse. The pathophysiological significance of L-carnitine implicates a potential role of the molecule in the management of varicocele, but the evidence so far is controversial for any recommendations. L-carnitine might be taken into consideration in selected cases; however, further search is needed in order for the optimal role of L-carnitine in infertile patients with varicocele to be clarified [37].

This study investigated the efficacy of an oral administration of a male fertility supplement containing substances known to relieve oxidative stress (Proxeed Plus; Sigma-Tau) in men with diagnosed idiopathic or varicocele-related infertility. General semen parameters were evaluated in accordance with the WHO reference values (5th edition): the ejaculate volume, concentration, and total number of spermatozoa in the ejaculate, and their motility and morphology (the percentage of forms with pathological morphology) at the time of diagnosis and after 3 and 6 months of treatment [2]. To the best of our knowledge, there have been no previous studies on the efficacy of Proxeed Plus in infertile men, particularly those diagnosed with varicocele. Its use in patients in this group is justified as these men have elevated levels of compounds with an oxidative potential, i.e., malondialdehyde (MDA) and nitric oxide (NO), and the decreased activity of enzymatic (superoxide dismutase [SOD], glutathione peroxidase [GPX]) and non-enzymatic antioxidants (ascorbic acid) in their semen [38]. All subjects in the present study showed a significant reduction in the value of at least one of the parameters relative to the lower limits of the WHO reference range. These observations are consistent with reports from many other research groups [15,39]. The use of the antioxidant preparation led to significant improvements in the absolute total sperm count in the semen (81%) as well as the sperm concentration (85%) and progressive (65%) and total (55%) motility. The detailed analysis of the sperm motility indicated significant increases in the percentage, concentration, and total number of motile sperm forms (A, B, and C forms) in at least 50% of the men in the study. However, there was no significant improvement in the sperm morphology, with improvement seen in only 22% of the subjects. Our data are consistent with previous reports of antioxidant therapy in patients with infertility of different aetiologies and varicocele-related infertility [14,18]. The efficacy of the preparation tested here, Proxeed Plus (Sigma-Tau), appeared to be due to the cumulative action of its individual antioxidant components.

The major component of the tested preparation was L-carnitine, which was first reported to have a positive influence on the semen quality by Tanphaichitr (1977), who found that the addition of acetyl-L-carnitine to ejaculate significantly improved the motility of spermatozoa [40]. Later, L-carnitine was also shown to play an important role in the process of energy production at the cellular level by participating in the beta-oxidation of long-chain fatty acids [41]. Its significant antioxidant potential is also an important property. The use of L-carnitine, either individually or in combination with other substances, such as acetyl-L-carnitine, as in the present study, has been shown to contribute to increases in the total sperm count, sperm concentration, and motility in subjects previously diagnosed with infertility due to a variety of aetiological factors. However, in contrast to our results, some studies have also reported significant improvements in sperm morphology with antioxidant therapy [14,18,23]. It is possible that such significant changes would have also been observed in the present study if the patient population had been larger. The pathophysiological significance of L-carnitine implicates a potential role of the molecule in the management of varicocele, but the evidence so far is controversial for any recommendations. L-carnitine might be taken into consideration in selected cases; however, further research is needed in order for the optimal role of L-carnitine in infertile patients with varicocele to be clarified [37]. Minerals, in particular zinc and selenium, seem to play key roles in regulating the antioxidant potential of male sperm. Zinc is an essential element for the proper functioning of SOD, while selenium is essential for GPX, both of which are important components of the antioxidant protective system of sperm [42]. Selenium deficiency may contribute not only to increased oxidative stress but also to reduced testicular size, sperm duct atrophy, and structural defects during sperm maturation in the epididymis, while zinc deficiency may contribute to adverse changes to DNA damage repair and cell division [8,42]. Selenium deficiency is common in men with varicocele, and reduced levels of selenium are directly correlate with lower sperm counts in semen as well as poorer motility and morphology [8,42]. Selenium supplementation has been shown to significantly reduce the peroxidation of sperm cell membranes and DNA and to improve the general semen parameters, particularly sperm motility and viability [8,13]. Similarly, zinc concentration is directly correlated with the total number of sperm in semen [8,42] and inversely proportional with the degree of sperm cell DNA fragmentation [5]. Zinc is a component of many metalloenzymes, including SOD, which is responsible for reducing oxidative stress, and many others involved in transcription, translation, and replication [8,13,38,43,44]. Through its cofactor, antiapoptotic, and antioxidant functions, zinc enables the maintenance of an adequate sperm motility and the progression of spermatogenesis [38,42,43,44,45]. Due to its antioxidant potential and its role as part of the mitochondrial oxidative respiratory chain, coenzyme Q10 appears to play a role not only in building the antioxidant potential but also in the process of energy production for sperm motility [14,46,47]. The treatment with coenzyme Q10 improves the general semen parameters, including increases in the sperm concentration and progressive motility [14,47,48,49]. Folic acid is important in DNA biosynthesis, which is highly active in germ cells. In addition, it has strong antioxidant properties through which it can contribute to reducing the oxidative stress and preventing the peroxidation of the cell membrane lipids [38,45]. Ascorbic acid (vitamin C) is also an antioxidant with a high antioxidant potential. A low ascorbic acid concentration in semen leads to DNA fragmentation and the deterioration of other parameters of semen quality [18,50]. Its presence in seminal fluid significantly protects spermatozoa from DNA peroxidation and promotes an improved motility [8,51].

There were no significant differences in the general semen quality parameters between the idiopathic infertility and varicocele-related infertility groups before the treatment, except in progressive motility, which although within the reference range in both groups, was significantly worse in patients with varicocele than in the idiopathic infertility group. This was most likely due to the inclusion of patients with Grade III varicocele, characterized not only by a significantly lower total sperm number in semen compared to men with idiopathic infertility, but also to their significantly reduced progressive motility resulting from the significantly lower numbers of form A spermatozoa in semen. Although there were no significant differences in other general semen parameters between the two groups, it is noteworthy that the varicocele-related infertility group tended to have lower mean and median values of almost all parameters than the idiopathic infertility group. These differences were likely due to higher levels of oxidative stress and reduced activity of enzymatic and non-enzymatic antioxidants in semen in the former, which has indeed been reported previously in patients with varicocele [15]. As a result of existing pathologies, both groups showed significant decreases in the total sperm count and concentration relative to the lower limit of the WHO reference range, consistent with previous studies. On the other hand, the detailed analyses of sperm motility showed no significant differences in the percentage, concentration, or total numbers of A, B, C, and D forms between the two groups. After antioxidant therapy, both groups showed an improvement in the semen quality with increases in the total number of spermatozoa in the ejaculate, sperm concentration, as well as the total and progressive motility. The magnitudes of improvement in all parameters were not significantly different between the groups and there were no differences in the effects of the treatment between the groups, confirming that it had a positive effect on the general semen parameters independent of the underlying etiology of infertility.

A detailed analysis taking into account the varicocele grade revealed significantly poorer parameters with an increasing severity of pathology. Patients with Grade III varicocele showed significantly worse results compared to Grade I. These results are in line with previous studies [14,15,23,52,53,54,55]. After a pharmacological intervention, patients with Grade III varicocele showed a significant improvement in their outcome compared to groups with lower varicocele grades, particularly Grade II in terms of the mean improvement after therapy, which may have been a direct result of the significant increase in the percentage of B form sperm with a slow progressive motility in this subgroup.

Currently, there is still a lack of defined guidelines for the treatment of male infertility associated with oxidative stress, partly due to the unknown etiology of the condition [9]. Over the past few years, several clinical trials have been conducted to investigate the effects of antioxidant supplementation on oxidative stress in seminal fluid and semen parameters [9,56]. Many have reported promising effects of antioxidants on sperm concentration, motility, morphology, and DNA fragmentation [57,58]. Some of the studies have shown improvements in the sperm redox status and semen parameters and a good correlation with pregnancy outcomes [59]. However, the role of antioxidant therapy in male infertility is still controversial. Countered by randomized studies that showed no improvement in the semen parameters and DNA fragmentation in infertile men, and no beneficial effect on pregnancy or live birth rates was observed. [60].

As it has been established that smoking can have an adverse effect on the quality of semen [61], the similar percentage of smokers in both groups was also an important advantage, which makes it possible to exclude this as a possible factor responsible for the observed differences between the groups. However, as data in the literature indicate no differences in the semen quality between nonsmokers and ex-smokers, we did not subdivide our study population according to this criterion, which may represent a limitation of the analysis and should be analyzed in future studies on a larger population. Similarly, the lack of history taking regarding passive smoking or the number of cigarettes smoked per day by active smokers also represents a limitation, as the distribution of highly dependent smokers between the two groups was not known. Information on the number of successful pregnancies after the treatment was also not collected.

## 5. Conclusions

The treatment of men with idiopathic or varicocele-related infertility with a supplement containing l-carnitine (Proxeed Plus; Sigma-Tau) resulted in significant improvements in the general semen parameters, particularly sperm count, sperm concentration, total motility, and progressive motility. This antioxidant preparation showed a particularly marked therapeutic benefit in patients with Grade III varicocele, with a greater improvement in progressive motility than in men with less severe or no varicocele. However, according to the literature, surgical treatment produces the best results in such patients [14,23,31,38]. The use of the antioxidant preparation examined here seems reasonable in men with idiopathic infertility and as an adjuvant in those with varicocele-related infertility in whom surgical treatment has resulted in no improvement. Its use should be considered particularly in patients with Grade III varicocele who do not wish to undergo surgical treatment or in whom such treatment is not possible for various reasons. A regular use of the preparation for a period of at least 3 months may lead to a significant improvement in the general semen parameters and make it easier to carry out in vitro fertilization procedures in couples in which male factors are responsible for infertility.

## Figures and Tables

**Table 1 jcm-11-06391-t001:** Interpretation of the maximal changes observed in sperm parameters after treatment compared to baseline values.

Parameters	Increase		No Change		Decrease		Test Result	*p* *
	*n*	%	*n*	%	*n*	%		
Volume of ejaculate, mL	69	42%	37	23%	57	35%	13.054	0.001
Sperm concentration, million/mL	139	85%	14	9%	10	6%	123.705	<0.001
Total sperm count, million/ejaculate	132	81%	10	6%	21	13%	97.425	0.001
Pathological forms, %	36	22%	112	69%	15	9%	2.443	0.294
Total motility, %	90	55%	59	36%	14	9%	29.711	<0.001
Progressive motility, %	107	66%	32	20%	24	15%	48.088	<0.001

* Friedman test. Data represent the greatest changes in parameters at 6 months compared to baseline.

**Table 2 jcm-11-06391-t002:** Effects of treatment on percentage of sperm motility forms in the ejaculate.

Parameters	Increase		No Change		Decrease		Test Result	*p **
	*n*	%	*n*	%	*n*	%		
Form A	88	54%	40	25%	35	21%	7.254	0.027
Form B	120	74%	14	9%	29	18%	46.466	<0.001
Form C	82	50%	30	18%	51	31%	19.642	<0.001
Form D	60	37%	42	26%	61	37%	1.568	0.456

* Friedman test.

**Table 3 jcm-11-06391-t003:** Effects of treatment on concentration of sperm motility forms in the ejaculate.

Parameters	Increase		No Change		Decrease		Test Result	*p **
	*n*	%	*n*	%	*n*	%		
Form A	108	66%	49	30%	6	4%	86.196	<0.001
Form B	126	77%	28	17%	9	6%	110.689	<0.001
Form C	88	54%	44	27%	31	19%	8.682	0.013
Form D	74	45%	54	33%	35	21%	0.0629	0.969

* Friedman test.

**Table 4 jcm-11-06391-t004:** Effects of treatment on total counts of sperm motility forms in the ejaculate.

Parameters	Increase		No Change		Decrease		Test Result	*p **
	*n*	%	*n*	%	*n*	%		
Form A	116	71%	36	22%	11	7%	81.094	<0.001
Form B	130	80%	17	10%	16	10%	96.768	<0.001
Form C	99	61%	32	20%	32	20%	12.931	0.002
Form D	76	47%	41	25%	46	28%	2.151	0.341

* Friedman test.

**Table 5 jcm-11-06391-t005:** Effects of treatment on general sperm parameters in relation to the presence of *varicocele*.

Parameter	*Varicocele*	Baseline					U Test	*p **	Change after Therapy					U Test	*p **
		$\bar{x}$ Mean	SD	Min	Median	Max			$\bar{x}$ Mean	SD	Min	Median	Max		
Volume of ejaculate, mL	No	3.7	1.0	2.0	3.5	6.0	1.508	0.132	0.0	0.8	–1.8	0.0	1.8	0.318	0.750
	Yes	3.4	1.0	1.5	3.5	6.0			0.0	0.9	–2.5	0.0	2.8		
Sperm concentration, million/mL	No	6.9	10.6	0.0	5.5	91.0	0.680	0.496	2.6	3.2	–9.0	2.5	18.0	1.104	0.270
	Yes	6.8	9.2	0.0	4.5	55.0			2.2	4.0	–19.0	2.0	24.0		
Total sperm count, million/ejaculate	No	25.4	47.3	0.0	18.2	409.5	1.020	0.308	8.7	11.0	–28.5	8.0	49.6	1.871	0.061
	Yes	23.0	31.3	0.0	15.0	192.5			6.0	10.2	–24.5	3.5	34.3		
Pathological forms, %	No	96.6	1.2	94.0	96.0	99.9	0.589	0.556	3.8	19.0	–2.0	0.0	99.9	–1.786	0.074
	Yes	96.5	1.6	93.0	96.0	99.9			4.8	21.0	–1.5	0.0	100.0		
Total motility, %	No	72.6	36.0	0.0	90.5	100.0	1.553	0.120	10.8	20.0	–22.2	0.7	100.0	–1.230	0.219
	Yes	67.1	32.2	0.0	77.8	100.0			15.1	22.9	–33.3	8.3	100.0		
Progressive motility, %	No	53.8	37.2	0.0	57.1	100.0	2.072	0.038	11.4	19.2	–30.0	7.2	100.0	–1.459	0.145
	Yes	40.8	33.2	0.0	33.3	100.0			16.2	22.0	–33.3	12.5	75.0		

* Mann–Whitney U test.

**Table 6 jcm-11-06391-t006:** Effects of treatment on percentages of sperm motility forms in the ejaculate in relation to the presence of *varicocele*.

Parameter	*Varicocele*	Baseline					U Test	*p **	Change after Therapy					U Test	*p **
		$\bar{x}$ Mean	SD	Min	Median	Max			$\bar{x}$ Mean	SD	Min	Median	Max		
Form A	No	28.0	33.7	0.0	16.2	100.0	1.143	0.253	7.6	18.6	–37.5	0.0	66.7	–0.800	0.424
	Yes	20.2	27.4	0.0	11.1	100.0			8.1	18.5	–50.0	4.2	50.0		
Form B	No	25.8	30.9	0.0	16.7	100.0	0.436	0.663	10.8	22.0	–50.0	10.3	100.0	–1.112	0.266
	Yes	20.7	24.3	0.0	14.3	100.0			13.3	20.8	–44.4	14.3	66.7		
Form C	No	18.8	23.2	0.0	13.4	75.0	–1.893	0.058	3.9	20.9	–40.0	0.0	100.0	–0.588	0.556
	Yes	26.2	26.6	0.0	22.2	100.0			5.4	20.1	–46.7	4.4	45.5		
Form D	No	21.0	31.0	0.0	0.0	100.0	–1.757	0.079	1.6	17.4	–33.3	0.0	100.0	–0.072	0.943
	Yes	24.7	26.1	0.0	18.2	100.0			2.6	25.2	–55.6	0.0	100.0		

* Mann–Whitney U test.

**Table 7 jcm-11-06391-t007:** Effects of treatment on concentrations of sperm motility forms in the ejaculate in relation to the presence of ***varicocele***.

Parameter	*Varicocele*	Baseline					U Test	*p **	Change after Therapy					U Test	*p **
		$\bar{x}$ Mean	SD	Min	Median	Max			$\bar{x}$ Mean	SD	Min	Median	Max		
Form A	No	1.4	1.9	0.0	1.0	8.0	0.972	0.331	1.2	1.6	–1.0	1.0	8.0	0.456	0.648
	Yes	1.1	1.6	0.0	0.1	6.0			1.0	1.3	–2.0	0.5	5.0		
Form B	No	1.4	1.7	0.0	1.0	6.0	0.338	0.735	1.4	1.8	–2.0	1.0	11.0	–0.057	0.955
	Yes	1.2	1.6	0.0	1.0	8.0			1.4	1.5	–1.0	1.0	7.0		
Form C	No	1.5	2.3	0.0	1.0	9.0	–0.917	0.359	0.7	2.4	–8.0	0.1	14.0	0.334	0.738
	Yes	1.8	2.7	0.0	0.3	12.0			0.5	1.7	–5.0	0.1	10.0		
Form D	No	2.5	8.6	0.0	0.0	73.0	–1.190	0.234	0.3	1.7	–5.0	0.0	8.0	0.063	0.950
	Yes	2.7	6.7	0.0	0.5	47.0			0.0	2.7	–20.0	0.0	9.0		

* Mann–Whitney U test.

**Table 8 jcm-11-06391-t008:** Effects of treatment on total counts of sperm motility forms in the ejaculate in relation to the presence of ***varicocele***.

Parameter	*Varicocele*	Baseline					U test	*p **	Change after therapy					U test	*p **
		$\bar{x}$ Mean	SD	Min	Median	Max			$\bar{x}$ Mean	SD	Min	Median	Max		
Form A	No	5.0	6.6	0.0	3.2	31.5	1.126	0.260	4.4	7.0	–5.8	3.3	43.5	0.616	0.538
	Yes	3.7	5.2	0.0	0.5	21.0			3.5	5.0	–9.0	2.0	22.3		
Form B	No	5.1	6.5	0.0	2.8	24.0	0.275	0.783	4.8	6.8	–7.0	3.0	42.0	0.436	0.663
	Yes	4.3	6.1	0.0	2.5	36.0			4.2	5.4	–9.0	2.2	22.3		
Form C	No	5.6	9.0	0.0	2.0	42.8	–0.837	0.402	2.3	9.1	–27.5	0.6	63.5	–0.249	0.804
	Yes	5.8	8.7	0.0	1.1	37.5			1.8	5.3	–16.5	0.8	25.5		
Form D	No	9.8	37.9	0.0	0.0	328.5	–0.996	0.319	–0.1	15.6	–118.5	0.0	48.5	0.065	0.948
	Yes	9.3	23.7	0.0	1.5	164.5			–1.3	9.0	–43.0	0.3	10.3		

* Mann–Whitney U test.

## Data Availability

The data presented in this study are available on request from the corresponding author.

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
