# Peer review of "The Role and Place of Antioxidants in the Treatment of Male Infertility Caused by Varicocele"

_jcm, 2022, doi:10.3390/jcm11216391_

Round 1
Reviewer 1 Report
In the manuscript titled “The role and place of antioxidants in the treatment of male infertility caused by varicocele,” the authors distinguish the role and application of Proxeed Plus as an antioxidant medication to male infertility treatment induced by varicocele.
1- I suggest changing the title of the study in order to match the experiment, for example: the effect of Proxeed Plus in the semen parameters of the men with varicocele
2- The abstract does not include the conclusion
3- The introduction is too long, there is some information about the varicocele which is not necessary. Please modify accordingly.
4- It is better if the authors explain the supplement in detail.
5- The aim and hypothesis of the study have been missed.
6- There is a lack in highlighting gaps in current understanding or conflicts in current knowledge about the subject of the manuscript in the introduction. Also, the need for investigations in the topic area is not demonstrated. Please include.
7- Is there any specific reason to use the data back to 2017? The authors could follow up for more than 6 months.
8- In the material- method, it’s better to talk about the semen parameters, which method did they use and based on which guideline?
9- The authors did not consider the effect of other factors such as age, obesity, smoking, alcohol drinking, exercise, etc.
10- The abstinence time can affect semen parameters. The authors should add this information to the study.
11- Table 1, the legend needs to be corrected and it is better to present data as before and after treatment.
12- The number of tables mixed up. Please correct them.
13- For the motility criteria the authors used unfamiliar expressions such as local motility. Please explain it and add the reference for it.
14- For table 1 in page 6, what do the authors mean by: concentration and count of sperm motility. Motility is considered based on percentage.
15- The authors mentioned they used 2 groups, one with idiopathic infertility and the other with varicocele. Although they did not compare or present any data about the idiopathic group.
16- Adding a post varicocele repair will add more wight to the results.
17- Discussion is too long (3 pages) and lacks depth (it is purely descriptive). Please remove unnecessary redundancies.
Author Response
We sincerely thank the Reviewers for valuable and insightful comments.
Reviewer #1:
Comment 1: I suggest changing the title of the study in order to match the experiment, for example: the effect of Proxeed Plus in the semen parameters of the men with varicocele.
Response: We appreciate the reviewer’s comment. There are many compounds containing antioxidants on the market with similar composition and we would not like the title to suggest the use of any of them, the study was not sponsored by a pharmaceutical company or any drug marketing agency, so the term "antioxidants" seems more appropriate to us.
Comment 2: The abstract does not include the conclusion.
Response: We thank the reviewer for this comment. We have added following sentences to the abstract:
“The use of the antioxidant preparation examined here seems reasonable in men with idiopathic infertility and as an adjuvant in those with varicocele-related infertility in whom surgical treatment has resulted in no improvement. Its use should be considered particularly in patients with Grade III varicocele who do not wish to undergo surgical treatment or in whom such treatment is not possible for various reasons.”
Comment 3: The introduction is too long, there is some information about the varicocele which is not necessary. Please modify accordingly.
Response: We thank the reviewer for this comment. We have shortened the introduction.
Comment 4: It is better if the authors explain the supplement in detail.
Response: We appreciate the reviewer’s comment. We have added more information about supplement to the manuscript:
“All patients received treatment with an antioxidant male fertility supplement containing L-carnitine fumarate (1725 mg), acetyl-L-carnitine (500 mg), ascorbic acid (vitamin C, 90 mg), coenzyme Q10 (20 mg), zinc (10 mg), folic acid (200 μg), selenium (50 μg), and vitamin B12 (1.5 μg) (Proxeed Plus; Sigma-Tau, Pomezia, Rome, Italy) twice a day for a period of 6 months from the time of the diagnosis of infertility.”
Comment 5: The aim and hypothesis of the study have been missed.
Response: Some scientific papers indicate that oral intake of antioxidants improves semen parameters. In our clinical practice, we used a compound containing antioxidants and performed a semiological evaluation after 6 months of therapy (two completed cycles of spermatogenesis). In this study, we present clinical results from our clinic.
Comment 6: There is a lack in highlighting gaps in current understanding or conflicts in current knowledge about the subject of the manuscript in the introduction. Also, the need for investigations in the topic area is not demonstrated. Please include.
Response: We appreciate the reviewer’s comment. We have added following sentences to the manuscript:
“The rationale behind oral antioxidants intake and positive effects on male reproduction outcome is only supported by few studies [31]. “
“Although there are some contrasting reports, oral consumption of compounds with antioxidant activity appears to improve sperm parameters, such as motility and concentration, and decrease DNA damage, but there is not sufficient evidence that fertility rates and live birth really improve after antioxidants intake. Moreover, it depends on the type of antioxidants, treatment duration, and even the diagnostics of the man’s fertility, among other factors [31]. In this study we present our clinical results.”
Comment 7: Is there any specific reason to use the data back to 2017? The authors could follow up for more than 6 months.
Response: We assumed that two cycles of spermatogenesis each for up to 74 days would be sufficient time to evaluate the results of antioxidant therapy. In 2017, we began to standardize the data, then analyzed the clinical conditions for including individual patients in the study. Due to the still ongoing scientific discussion on this topic, we decided to present our results collected by the andrological department of the clinic. In addition, changes in the 6th manual compared to the 5th edition do not affect the evaluation of the analyzed parameters in this paper, which does not change its validity.
Comment 8: In the material- method, it’s better to talk about the semen parameters, which method did they use and based on which guideline?
Response: We thank the reviewer for this comment. We have added following sentences to the manuscript:
“Basic seminal parameters were evaluated by a European Society of Human Repro-duction and Embryology (ESHRE) -certified embryologist following the fifth edition of the World Health Organisation (2010) guidelines.”
“Semen, after 2-5 days of sexual abstinence, was donated by patients into sterile containers. Semen analysis took place using a Makler chamber. A Makler sperm counting chamber (Sefi Medical Instruments, Haifa, Israel) inserted into a light microscope (Carl Zeiss Jena, Jena, Germany) with Ph2 phase contrast was used for microscopic evaluation of semen. A small drop of semen, well mixed by pipetting, was placed in the center of a Makler chamber, and the chamber was covered with a coverslip. The preparations were then viewed under a magnification of 200× at room temperature. The type of movement was then assessed, distinguishing among fast (type A) and slow (type B) forward (type A) and slow (type B) sperm, non-progressive (type C) sperm (local motility) and non-moving (type D) sperm (no visible motility).”
Comment 9: The authors did not consider the effect of other factors such as age, obesity, smoking, alcohol drinking, exercise, etc.
Response: The group of patients with idiopathic infertility, on physical examination and subject examination, did not show any potential causes of infertility that are commonly known. This included environmental, lifestyle hygiene factors. In the group of patients with varicocele, we excluded patients who showed any potential causes of infertility in their history and in laboratory tests and physical examination. During treatment, other than taking the product, we did not follow any additional recommendations for lifestyle changes.
Comment 10: The abstinence time can affect semen parameters. The authors should add this information to the study.
Response: Patients maintained a 2-5 days abstinence period according to 5th manual standards. The period of abstinence at each examination was within this range.
We have added following sentence to the manuscript for clarification:
“Semen, after 2-5 days of sexual abstinence, was donated by patients into sterile containers.”
Comment 11: Table 1, the legend needs to be corrected and it is better to present data as before and after treatment.
Response: We appreciate the reviewer’s comment. We have corrected the legend:
“Interpretation of the maximal changes observed in sperm parameters after treatment in compare to baseline values.”
Pre- and post-treatment data are presented in Tables 3 and 4.
Comment 12: The number of tables mixed up. Please correct them.
Response: Corrected.
Comment 13: For the motility criteria the authors used unfamiliar expressions such as local motility. Please explain it and add the reference for it
.
Response: Local motility refers to category C spermatozoa not showing progressive movement but showing local movement - not progressive.
The following sentences are included in the manuscript:
“…The type of movement was then assessed, distinguishing among fast (type A) and slow (type B) forward (type A) and slow (type B) sperm, non-progressive (type C) sperm (local motility) and non-moving (type D) sperm (no visible motility).”
“Analysis of sperm motility (A and B forms, fast and slow progressive motility, re-spectively; C, local motility; D, no visible motility)…”
Comment 14: For table 1 in page 6, what do the authors mean by: concentration and count of sperm motility. Motility is considered based on percentage.
Response: Concentration - number of sperm per unit volume (milliliter) of semen.
Count of sperm motility - the absolute number of A, B, C and D sperm forms present in the ejaculate.
Comment 15: The authors mentioned they used 2 groups, one with idiopathic infertility and the other with varicocele. Although they did not compare or present any data about the idiopathic group.
Response: We thank the Reviewer for this comment. Idiopathic group is non varicocele group.
Comment 16: Adding a post varicocele repair will add more weight to the results.
Response: Patients qualified for the study did not undergo surgery for various reasons. Most frequently they did not consent to surgery or had contraindications for the varicocele repair.
Comment 17: Discussion is too long (3 pages) and lacks depth (it is purely descriptive). Please remove unnecessary redundancies.
Response: We thank the Reviewer for this comment. We have shortened the discussion.

Reviewer 2 Report
Authors should be commended on an interesting study. I have several questions/comments:
1. In the disclosures, it states that this study was not funded. This should be stated more clearly in the methods section, as otherwise this study could be construed as having been funded by the drug company since this uses a name-brand drug.
2. How was this particular medication chosen, and how was the dosing treatment determined? Please report if any negative side effects on the medication were noted.
3. Did the results control for semen analysis abstinence interval? This would be important to control for, as abstinence interval can significantly affect motility parameters.
4. Some of the tables are difficult to read, and would consider presenting this data in graphical pre-treatment and post-treatment format.
I appreciate that the authors acknowledge the limitations including absence of pregnancy and live birth data, as that is the clinically significant outcome. Would be interesting to note if these changes in semen parameters resulted in clinically significant changes in outcomes, such as decreased need for ART or increased rates of spontaneous conception, or time from completion of treatment to time of conception.
Author Response
We sincerely thank the Reviewers for valuable and insightful comments.
Reviewer #2:
Comment 1: In the disclosures, it states that this study was not funded. This should be stated more clearly in the methods section, as otherwise this study could be construed as having been funded by the drug company since this uses a name-brand drug
Response: We thank the reviewer for this comment. The study was not funded by a pharmaceutical company or any other organization.
We have added following sentence:
“The study was not sponsored by a pharmaceutical company or any drug marketing agency.”
Comment 2: How was this particular medication chosen, and how was the dosing treatment determined? Please report if any negative side effects on the medication were noted.
Response: Each patient received Proxeed Plus therapy because we found it to be the one of the most versatile and adequate medication. Each patient was given 2 x 1 doses. There were no significant side effects of medications. There were no cases where treatment had to be discontinued due to side effects.
Comment 3: Did the results control for semen analysis abstinence interval? This would be important to control for, as abstinence interval can significantly affect motility parameters.
Response: We thank the reviewer for this comment. Patients maintained a 2-5 day abstinence period according to guidelines. The period of abstinence at each examination was within this range.
Comment 4: Some of the tables are difficult to read, and would consider presenting this data in graphical pre-treatment and post-treatment format.
Response: We have prepared many versions of graphical charts, unfortunately none of them was able to contain all the information. If necessary, we will gladly provide graphic charts as supplementary material.
Minor Comment 1: I appreciate that the authors acknowledge the limitations including absence of pregnancy and live birth data, as that is the clinically significant outcome. Would be interesting to note if these changes in semen parameters resulted in clinically significant changes in outcomes, such as decreased need for ART or increased rates of spontaneous conception, or time from completion of treatment to time of conception.
Response: We appreciate the reviewer’s comment. We had intended to do so, unfortunately most of the patients participating in the study were under the care of our clinic's andrology department. A significant number of patients were no longer in contact with us after treatment, therefore we were unable to collect data on their follow-up.
Round 2
Reviewer 1 Report
Please change the term local motility to non-progressive motility which is a standard term.
Reviewer 2 Report
agree with acceptance for publication